# Stochastic Antiresonance for Systems with Multiplicative Noise and Sector-Type Nonlinearities

**DOI:** 10.3390/e26020115

**Published:** 2024-01-26

**Authors:** Adrian-Mihail Stoica, Isaac Yaesh

**Affiliations:** 1Faculty of Aerospace Engineering, University Politehnica of Bucharest, 060042 Bucharest, Romania; 2Control Department, Elbit Systems, Ramat-Hasharon 3100401, Israel; isaac.yaes@elbitsystems.com

**Keywords:** stochastic antiresonance, sector-bounded nonlinearities, stochastic systems with state- dependent noise, stability analysis, infinitesimal generator

## Abstract

The paradigm of stochastic antiresonance is considered for a class of nonlinear systems with sector bounded nonlinearities. Such systems arise in a variety of situations such as in engineering applications, in physics, in biology, and in systems with more general nonlinearities, approximated by a wide neural network of a single hidden layer, such as the error equation of Hopfield networks with respect to equilibria or visuo-motor tasks. It is shown that driving such systems with a certain amount of state-multiplicative noise, one can stabilize noise-free unstable systems. Linear-Matrix-Inequality-based stabilization conditions are derived, utilizing a novel non-quadratic Lyapunov functional and a numerical example where state-multiplicative noise stabilizes a nonlinear system exhibiting chaotic behavior is demonstrated.

## 1. Introduction

Stochastic Antiresonance (SAR) is an intriguing paradigm, wherein nonlinear dynamic systems that are unstable or just marginally so can be stabilized by driving them with state-multiplicative noise. The reverse phenomenon, Stochastic Resonance (SR), has been studied in the context of periodic occurrences of ice ages [1]. A few examples of SR were described and analyzed in [1,2], such as particle in a double well, animal behavior, sensory neurons and ionic channels in biological cells, optical systems, electronic devices, and so on. In one of these studies [2], SAR in laser systems was also analyzed. All these examples exhibit a bell-shaped Signal-to-Noise Ratio (SNR) as a function of the applied noise level. The action of SAR has also been studied (e.g., [3]) in the context of squid giant axons and the potential for therapeutic neurological applications was pointed out. Other antiresonance applications may also be found, for instance, in electronics and in mechanical engineering (e.g., [4,5,6]) for the vibration-suppression of certain components of the system. One may even consider the option of applying SAR as an alternative to deterministic adaptive controllers for neuromodulation (e.g., [7]). While many such paradigms can be analyzed using numerical simulations of the relevant dynamic models, the development of stochastic stability analysis tools may contribute to the reliability in applying SAR. Examples of such models are Hopfield networks [8], which are symmetric recurrent neural networks that exhibit motions in the state space converging to minima of energy. Such networks are mentioned in the context of practical complex problems such as the implementation of associative memory, linear programming, and so on, and also play an important role in understanding human motor tasks involving visual feedback (e.g., [9,10]).

The aim of the present paper, is to analyze the SAR phenomenon for a class of systems with sector-bounded nonlinearities, in order to explore the potential application of state-multiplicative noise to control and stabilize such systems. The considered type of nonlinearities have been widely used in the context of the absolute stability concept introduced by Lur’e [11] and further studied in, for instance [12,13,14], to mention only a few major developments. The interest in sector-bounded nonlinearities is due to the wide area of practical applications in which the physical plants include control saturations, modeling uncertainties, time-delays, and measurements quantizations.

The main results presented in this paper provide conditions for the occurrence of SAR, and they are derived using specific methods for the stability analysis of stochastic nonlinear systems. The theoretical developments are illustrated by numerical examples.

Throughout the paper, $R^{n}$ denotes the *n* dimensional Euclidean space, $R^{n\times m}$ is the set of all n×m real matrices, and the notation X>0, (respectively, X≥0) for $X\in R^{n\times n}$ means that *X* is symmetric and positive definite (respectively, semi-definite). Tr{A} denotes the trace of the matrix *A* and λ(A) denotes its eigenvalue, whereas, $\bar{\lambda }\left(A\right)$ and $\underline{\lambda }\left(A\right)$, respectively, denote the maximum and minimum eigenvalues of a symmetric matrix *A*.

Furthermore, |w| for $w\in R^{n}$ will denote ${\left(w^{T}w\right)}^{\frac{1}{2}}$. Throughout the paper, (Ω,F,P) is a given probability space. Expectation is denoted by E{·}.

## 2. Preliminaries

In the present paper, we deal with stochastic systems that involve state-multiplicative noise represented using Itô type stochastic differential equations (SDEs) as follows:(1)$$\begin{array}{c} dx(t)=f(x(t),t)dt+g(x(t),t)d\beta (t), \end{array}$$
where β(t) is a zero mean *r*-dimensional Wiener process adapted to an increasing family of $F_{t\geq 0}$ of σ-algebras $F_{t}\subset F$, with $E\{d\beta \left(t\right)d\beta^{T}\left(t\right)\}=Qdt$. The state vector $x\left(t\right)\in R^{n}$ and it is assumed that the functions f(x(t),t) and g(x(t),t) satisfy the existence conditions for a unique solution of the above stochastic differential equation (see, e.g., [15,16,17]). For an initial condition $x_{0}$ at $t=t_{0}$ independent of the σ-algebra generated by β(t),t≥0, this solution will be denoted by $x(t,t_{0},x_{0})$. Assume that f(0,t)=0 and g(0,t)=0, ∀t≥0. Then, according to [18], the trivial solution x(t)≡0 of Equation (Equation 1) is called *stable in probability for t≥0* if for any $t_{0}\geq 0$ and ϵ≥0,
$$\begin{array}{c} \lim_{x_{0}\to 0}P\left\{\underset{t\geq 0}{\sup}\left|x\left(t,t_{0},x_{0}\right)\right|>\epsilon \right\}=0. \end{array}$$
Moreover, the solution x(t)≡0 is called asymptotically stable in probability if it is stable in probability and if
$$\begin{array}{c} \lim_{x_{0}\to 0}P\left\{\lim_{t\to \infty }x\left(t,t_{0},x_{0}\right)=0\right\}=1. \end{array}$$

One can prove (see, e.g., [18]) that x(t)≡0 is asymptotically stable in probability if there exists a twice continuously differentiable positive definite function V(x,t) such that LV<0, where the infinitesimal generator LV has the expression [16,17]
(2)$$\begin{array}{c} LV:=V_{t}+V_{x}^{T}f\left(x\left(t\right),t\right)+\frac{1}{2}Tr\left\{g\left(x\left(t\right),t\right)Qg^{T}\left(x\left(t\right),t\right)V_{xx}\right\}, \end{array}$$
in which $V_{t}$ and $V_{x}$ denote the first-order partial derivatives of V(x,t) with respect to *t* and *x*, respectively, and $V_{xx}$ is its second partial derivative with respect to *x*. This result represents a generalization of the well-known Lyapunov’s theorem on asymptotic stability from the deterministic framework. Although this type of a stability is weaker than mean square exponential stability (see, e.g., [19]), it is still of practical value as we will see in the sequel.

## 3. Motivation

Consider the following linear continuous-time scalar stochastic system [20]
(3)dx(t)=ax(t)dt+σx(t)dβ(t),
where a>0 and β(t) is a Wiener process with $E\{\beta^{2}\left(t\right)\}=dt$. This system is clearly unstable for σ=0. However, for σ≠0, we choose [18,21] the positive definite function $V\left(x\right)={\left|x\right|}^{\nu },\nu \in \left(0,1\right)$ and calculate
$$LV=V_{x}ax+\frac{1}{2}V_{xx}\sigma^{2}x^{2}={\left|x\right|}^{\nu }\nu \left(a+\frac{\sigma^{2}}{2}\left(\nu -1\right)\right).$$
Taking σ>0 such that $\sigma^{2}>2a$, it follows that for $\nu <1-\frac{2a}{\sigma^{2}}$, the infinitesimal generator LV<0 and, therefore, the origin is asymptotically stable in probability. We next see in Figure 1 a case of $\sigma^{2}<2a$ (Figure 1 top) and a case of $\sigma^{2}>2a$ (Figure 1 bottom).

The SAR phenomenon seems, at first sight, to be counter-intuitive. Indeed, it often happens, e.g., in stable systems, that multiplicative noise can even drive a system to instability. However, the reverse phenomenon of SAR is even more intriguing. A somewhat similar phenomenon is well known to control practitioners where dither (high-frequency periodical excitation) added to a control signal can eliminate limit cycles, by a sort of linearization, e.g., in the case of systems with dead-zone nonlinearities that are smoothed by the dither. In such cases, practitioners apply classical frequency-domain approximate analysis, i.e., describing-function-based analysis, in order to determine the dither characteristics, to eliminate the unwanted limit cycle. In SAR, the exciting noise is a white state-multiplicative noise rather than a periodic one, and the white multiplicative noise seems to have a balancing effect, which can push the system back towards the origin upon large deviations. In our example of System (Equation 3), the noise-free equilibrium is x=0; however, due to a>0, the system diverges. For non-zero *x*, the multiplicative noise term randomly provides corrections that either drive the system to zero, or away from zero, depending on the sign. Larger values of |x| result in larger convergence rates to the origin. However, once the state *x* passes through x=0, it stays there, since both the so-called drift term ax(t)dt and the diffusion term σx(t)dβ(t) are then nulled.

Our aim in the present paper is to demonstrate this intriguing phenomenon and provide an analysis tool that can determine an intensity of noise that can stabilize the system, such as the describing function does, in the case of dither.

## 4. Problem Formulation

Consider the following system: (4)$$\begin{array}{c} \begin{array}{ccc} dx(t) & = & Ax(t)dt+Ff(y(t))dt+Dx(t)d\beta (t)\\ y\left(t\right) & = & Cx\left(t\right)\\ x(0) & = & x_{0} \end{array} \end{array}$$
where $x\in R^{n}$ is the state vector, $y\in R^{n}$ is the measured system output, and $\beta \left(t\right)\in R$ is a standard Wiener process with $E\{\beta^{2}\left(t\right)\}=dt$ on the given probability space, which is also independent of $x_{0}$. The elements of *y* are $y_{i}=C_{i}x\in R,\;i=1,\ldots ,n$, where $C_{i}$ is the *i*’th row vector of *C*, namely $y_{i}=\sum_{j=1}^{n}C_{ij}x_{j}$, and the components $f_{i}\left(y_{i}\right)$ of f(y) satisfy the sector conditions $0\leq y_{i}f_{i}\left(y_{i}\right)\leq s_{i}y_{i}^{2}$ [11,22,23], which are equivalent to
(5)$$f_{i}\left(y_{i}\right)\left(f_{i}\left(y_{i}\right)-s_{i}y_{i}\right)\leq 0,\;i=1,\ldots ,n.$$
Let us define, for the sequel, $S=diag\{s_{1},s_{2},\ldots ,s_{n}\}$. We note that the model of System (Equation 4) is relevant also in cases where f(y(t)) in the model is not a priori sector-bounded. In such cases, one may invoke the universal approximation theorem [24] to systems where a single hidden layer, with, e.g., a tanh activation function and a linear output layer, provides an approximation with arbitrarily small error for an arbitrarily wide hidden layer. In such cases, the model of System (Equation 4) readily becomes relevant, as the approximate function is now sector-bounded. However, one should be very careful, as systems exhibiting chaotic behaviors may require a very high degree of approximation to maintain their chaotic nature.

## 5. Stability in Probability Analysis in the Absence of Nonlinearities

Consider first the simpler case where F=0 in System (Equation 4). The main result of this section is the following theorem.

**Theorem 1.** 
*If the following condition holds*

(6)
$$\begin{array}{c} \bar{\lambda }\left(A+A^{T}\right)<\sigma^{2}, \end{array}$$

*then the solution x(t)≡0 of System (Equation 4) with F=0 and $D=\sigma^{2}I$ is asymptotically stable in probability.*


**Proof.** Consider the positive definite function
$$V\left(x\right)={\left(x^{T}x\right)}^{\nu /2},\nu \in \left(0,1\right)$$
for which, one readily obtains that
$$V_{x}\left(x\right)=\nu {\left(x^{T}x\right)}^{\frac{\nu }{2}-1}x.$$
We next denote
(7)$$\begin{array}{c} \rho :=1-\nu /2\in (1/2,1) \end{array}$$
and obtain also
$$V_{xx}\left(x\right)=\nu \frac{1}{{\left(x^{T}x\right)}^{\rho }}\left(I-\frac{2\rho }{x^{T}x}xx^{T}\right).$$
Plugging $V_{x}$ and $V_{xx}$ into Expression (Equation 2) of the infinitesimal generator, we obtain
(8)$$\begin{array}{c} LV=V_{x}^{T}\left(x\right)Ax+\frac{1}{2}Tr\left\{Dxx^{T}D^{T}V_{xx}\left(x\right)\right\}, \end{array}$$
and using the above expressions derived above for $V_{x}$ and $V_{xx}$, the results show that
$$\begin{array}{c} LV=\frac{\nu }{2{\left(x^{T}x\right)}^{\rho }}x^{T}\left(A+A^{T}+D^{T}D\right)x-\frac{2\rho \nu }{{\left(x^{T}x\right)}^{\rho +1}}x^{T}\left(Dxx^{T}D^{T}\right)x. \end{array}$$
If we focus on the case D=σI, we obtain that
$$LV=\frac{\nu }{2{\left(x^{T}x\right)}^{\rho }}x^{T}\left(A+A^{T}+\sigma^{2}I-2\rho \sigma^{2}I\right)x.$$
From Equation (Equation 7), it follows that 1−2ρ=ν−1 and, therefore,
(9)$$\begin{array}{c} LV=\frac{\nu }{2{\left(x^{T}x\right)}^{\rho }}x^{T}\left[A+A^{T}-\sigma^{2}\left(1-\nu \right)I\right]x. \end{array}$$
Taking into account the condition (Equation 6) from the statement, it follows that there exists a small enough ν∈(0,1) such that
$$\begin{array}{c} A+A^{T}-\sigma^{2}\left(1-\nu \right)I<0 \end{array}$$
and, therefore, from Equation (Equation 9) it follows that LV<0, concluding, thus, that the solution x(t)≡0 of System (Equation 4) with F=0 is asymptotically stable in probability. □

## 6. Stability in Probability Analysis in the Presence of Nonlinearities

We consider now the case of F≠0 in System (Equation 4). Assume that the following conditions are accomplished.

**Hypothesis 1 (H1).** The derivatives of the nonlinearities are bounded, namely, there exist $\delta_{i}>0$, i=1,⋯,n, such that $\frac{df_{i}\left(y_{i}\right)}{dy_{i}}<\delta_{i}$, and

**Hypothesis 2 (H2).** The matrix *C* satisfies the condition $C^{T}C=I$.

**Remark 1.** 
*The assumption $C^{T}C=I$ may be fulfilled if the matrix $C^{T}C$ is nonsingular, performing the similarity transformation $\hat{x}=\hat{T}x$ with $\hat{T}:=\Sigma U^{T}$, where Σ and U are obtained from the singular value decomposition $C^{T}C=U\Sigma^{2}U^{T}$. In the case when $C^{T}C$ is not invertible, one may add fictitious new outputs such that C becomes invertible. The effect of these new added outputs may be vanished setting $s_{i}=0$ for their corresponding indices i.*


Then, the following result provides asymptotic stability conditions for System (Equation 4).

**Theorem 2.** *Assume that the assumptions* **H1** *and*  **H2** *hold. If there exist ν∈(0,1), $\Lambda =diag(\lambda_{1},\cdots ,\lambda_{n})$, $\lambda_{i}\geq 0$, i=1,⋯,n, and $T=diag\left(\tau_{1},\cdots ,\tau_{n}\right),\;\tau_{i}\geq 0$, i=1,⋯,n, such that*
(10)$$\begin{array}{c} \left[\begin{array}{cc} N_{11}\left(\nu ,\Lambda \right) & N_{12}\left(\nu ,\Lambda ,T\right)\\ N_{12}^{T}\left(\nu ,\Lambda ,T\right) & N_{22}\left(\Lambda ,T\right) \end{array}\right]<0 \end{array}$$
*where $\Delta :=diag\left(\delta_{1},\cdots ,\delta_{n}\right)$ and*
(11)$$\begin{array}{c} \begin{array}{ccc} N_{11}\left(\nu ,\Lambda \right) & := & \nu \left[A^{T}+A-\sigma^{2}\left(1-\nu \right)I\right]+\sigma^{2}C^{T}\Lambda \Delta C\\ N_{12}\left(\nu ,\Lambda ,T\right) & := & \nu F+{\left[A-\frac{\sigma^{2}}{2}\left(1-\frac{\nu }{2}\right)I\right]}^{T}C^{T}\Lambda +SC^{T}T\\ N_{22}\left(\Lambda ,T\right) & := & -2T+\Lambda CF+F^{T}C^{T}\Lambda \end{array} \end{array}$$
*then the solution x(t)≡0 of the stochastic system, System (Equation 4), with D=σI is asymptotically stable in probability for any sector-type nonlinearities $f_{i}\left(y_{i}\right)$, satisfying the conditions $0\leq y_{i}f_{i}\left(y_{i}\right)\leq s_{i}y_{i}^{2}$ and $\frac{df_{i}\left(y_{i}\right)}{dy_{i}}<\delta_{i}$, i=1,⋯,n.*

**Proof.** Consider the positive definite function
(12)$$\begin{array}{c} V\left(x\right)={\left(x^{T}x\right)}^{\nu /2}+\Sigma_{k=1}^{n}\lambda_{k}\int_{0}^{y_{k}}s^{-2\rho }f_{k}\left(s\right)ds, \end{array}$$
with $\lambda_{k}\geq 0,\;k=1,\ldots ,n$, ν∈(0,1) and ρ defined in Equation (Equation 7).

Then, direct computations give that
$$\begin{array}{c} V_{x}\left(x\right)=\nu {\left(x^{T}x\right)}^{-\rho }x+{\left(x^{T}C^{T}Cx\right)}^{-\rho }C^{T}\Lambda f \end{array}$$
and
$$\begin{array}{c} \begin{array}{ccc} V_{xx}\left(x\right) & = & \frac{\nu }{{\left(x^{T}x\right)}^{\rho }}\left(I-\frac{2\rho }{x^{T}x}xx^{T}\right)-\rho {\left(x^{T}C^{T}Cx\right)}^{-\rho -1}C^{T}\Lambda fx^{T}C^{T}C\\ & & +{\left(x^{T}C^{T}Cx\right)}^{-\rho }C^{T}\Lambda f_{y}C \end{array} \end{array}$$
where Λ was defined in the statement and the following notations have been used
(13)$$\begin{array}{c} \begin{array}{c} f:={\left[f_{1},\cdots ,f_{n}\right]}^{T}\;\mathrm{and}\;f_{y}:=diag\left(\frac{df_{1}}{dy_{1}},\cdots ,\frac{df_{n}}{dy_{n}}\right). \end{array} \end{array}$$
Further, define
(14)$$\begin{array}{c} -F_{0}:=V_{x}^{T}\left(Ax+Ff\right)+\frac{1}{2}x^{T}D^{T}V_{xx}Dx \end{array}$$
and the nonlinearities constraints
(15)$$\begin{array}{c} -F_{i}:={\left(x^{T}x\right)}^{-\rho }f_{i}\left(y_{i}\right)\left(f_{i}\left(y_{i}\right)-s_{i}y_{i}\right)\leq 0,\; \end{array}$$
i=1,…,n. Then, in accordance with the S-procedure technique (see, e.g., [23]), the stability condition
$$\begin{array}{c} LV=V_{x}^{T}\left(Ax+Ff\right)+\frac{1}{2}x^{T}D^{T}V_{xx}Dx<0 \end{array}$$
is accomplished together with Constraints (Equation 15), if there exist $\tau_{1},\cdots ,\tau_{n}\geq 0$, such that
(16)$$\begin{array}{c} F_{0}-\sum_{i=1}^{n}\tau_{i}F_{i}>0. \end{array}$$
Using the expressions of $V_{x}\left(x\right)$ and of $V_{xx}\left(x\right)$ derived above, it follows that Equation (Equation 16) is equivalent to
(17)$$\begin{array}{c} \begin{array}{c} \left[\nu {\left(x^{T}x\right)}^{-\rho }x^{T}+{\left(x^{T}C^{T}Cx\right)}^{-\rho }f^{T}\Lambda C\right]\left(Ax+Ff\right)\\ +\frac{1}{2}x^{T}D^{T}\left[\nu {\left(x^{T}x\right)}^{-\rho }\left(I-\frac{2\rho }{x^{T}x}xx^{T}\right)\right.\\ \left.-\rho {\left(x^{T}C^{T}Cx\right)}^{-\rho -1}C^{T}\Lambda fx^{T}C^{T}C\right.\\ \left.+{\left(x^{T}C^{T}Cx\right)}^{-\rho }C^{T}\Lambda f_{y}C\right]Dx\\ -{\left(x^{T}x\right)}^{-\rho }\left(f^{T}Tf-\frac{1}{2}f^{T}TCSx-\frac{1}{2}x^{T}SC^{T}Tf\right)<0 \end{array} \end{array}$$
where T was defined in the statement.

Multiplying (Equation 17) by ${\left(x^{T}x\right)}^{\rho }$, one obtains for $C^{T}C=I$,
(18)$$\begin{array}{c} \begin{array}{c} \left(\nu x^{T}+f^{T}\Lambda C\right)\left(Ax+Ff\right)\\ +\frac{1}{2}x^{T}D^{T}\left[\nu \left(I-\frac{2\rho }{x^{T}x}xx^{T}\right)-\frac{\rho }{x^{T}x}C^{T}\Lambda fx^{T}+C^{T}\Lambda f_{y}C\right]Dx\\ -f^{T}Tf+\frac{1}{2}f^{T}TCSx+\frac{1}{2}x^{T}SC^{T}Tf<0. \end{array} \end{array}$$

Taking as in the previous case, D=σI, Inequality (Equation 18) becomes
(19)$$\begin{array}{c} \begin{array}{c} \left(\nu x^{T}+f^{T}\Lambda C\right)\left(Ax+Ff\right)+\frac{\nu \sigma^{2}}{2}\left(1-2\rho \right)x^{T}x\\ -\frac{1}{2}\sigma^{2}\rho x^{T}C^{T}\Lambda f+\frac{1}{2}\sigma^{2}x^{T}C^{T}\Lambda f_{y}Cx\\ -f^{T}Tf+\frac{1}{2}f^{T}TCSx+\frac{1}{2}x^{T}SC^{T}Tf<0, \end{array} \end{array}$$
which may be rewritten in the equivalent form
(20)$$\begin{array}{c} \left[\begin{array}{cc} x^{T} & f^{T} \end{array}\right]\left[\begin{array}{cc} M_{11} & M_{12}\\ {M_{12}}^{T} & M_{22} \end{array}\right]\left[\begin{array}{c} x\\ f \end{array}\right]<0, \end{array}$$
where the following notations have been introduced
$$\begin{array}{c} \begin{array}{c} M_{11}:=\frac{\nu }{2}\left(A^{T}+A\right)+\frac{\nu \sigma^{2}}{2}\left(1-2\rho \right)I+\frac{\sigma^{2}}{2}C^{T}\Lambda \Delta C\\ M_{12}:=\frac{\nu }{2}F+\frac{1}{2}A^{T}C^{T}\Lambda -\frac{\sigma^{2}\rho }{4}C^{T}\Lambda +\frac{1}{2}SC^{T}T\\ M_{22}:=-T+\frac{1}{2}\left(\Lambda CF+F^{T}C^{T}\Lambda \right). \end{array} \end{array}$$
Using the definition of Δ from the statement, it follows that if Condition (Equation 10) is accomplished, then Inequality (Equation 20) holds for any $[x^{T}\;\;f^{T}]\neq 0$. Thus, one concludes that LV<0 together with the sector constraints $f_{i}\left(y_{i}\right)\left(f_{i}\left(y_{i}\right)-s_{i}y_{i}\right)\leq 0$, i=1,…,n are fulfilled and, therefore, the solution x(t)≡0 of System (Equation 4) is asymptotically stable in probability. Thus, the proof ends. □

**Remark 2.** 
*The above result may be extended to the more general case of a non-scalar matrix D>0. Thus, using the fact that $\underline{\lambda }\left(D\right)x^{T}x\leq x^{T}Dx\leq \bar{\lambda }\left(D\right)x^{T}x$ and $-2x^{T}DC^{T}\Lambda f\leq x^{T}D^{2}x+f^{T}\Lambda CC^{T}\Lambda f$, one obtains that, if there exist ν∈(0,1), $\Lambda =diag(\lambda_{1},\cdots ,\lambda_{n})$, $\lambda_{i}\geq 0$, i=1,⋯,n and $T=diag\left(\tau_{1},\cdots ,\tau_{n}\right),\;\tau_{i}\geq 0$, i=1,⋯,n, such that*

$$\begin{array}{c} \left[\begin{array}{ccc} P_{11}\left(\nu ,\Lambda \right) & P_{12}\left(\nu ,\Lambda ,T\right) & 0\\ P_{12}^{T}\left(\nu ,\Lambda ,T\right) & P_{22}\left(\Lambda ,T\right) & P_{23}\left(\nu ,\Lambda \right)\\ 0 & P_{23}^{T}\left(\nu ,\Lambda \right) & -I \end{array}\right]<0, \end{array}$$

*where*

$$\begin{array}{c} \begin{array}{ccc} P_{11}\left(\nu ,\Lambda \right) & := & \nu \left[A^{T}+A+D^{2}-2\rho {\underline{\lambda }}^{2}\left(D\right)I\right]+\frac{\rho \bar{\lambda }\left(D\right)}{2}D^{2}+DC^{T}\Lambda \Delta CD\\ P_{12}\left(\nu ,\Lambda ,T\right) & := & \nu F+A^{T}C^{T}\Lambda +SC^{T}T\\ P_{22}\left(\Lambda ,T\right) & := & -2T+\Lambda CF+F^{T}C^{T}\Lambda \\ P_{23}\left(\nu ,\Lambda \right) & := & \sqrt{\frac{\rho \bar{\lambda }\left(D\right)}{2}}\Lambda C, \end{array} \end{array}$$

*with $\rho =1-\frac{\nu }{2}$, then the solution x(t)≡0 of the stochastic system, System (Equation 4), is asymptotically stable in probability. The proof is similar to that of Theorem 2 and, therefore, it is omitted.*


**Remark 3.** 
*Note that Conditions (Equation 10) and (Equation 11) of Theorem 2, are convex in the system matrices A,F and also in the noise intensity $\sigma^{2}$. These facts allow verification of the SAR condition not only in fixed values of A,F, and $\sigma^{2}$ but also within a convex hull of those parameters, allowing a merit beyond the one provided by numerical simulations. Note that the conservatism in the conditions of Theorem 2 can be reduced by expanding Condition (Equation 10) using Schur complements as*

$$\begin{array}{c} \left[\begin{array}{ccc} {\tilde{N}}_{11} & N_{12} & N_{13}\\ N_{12}^{T} & N_{22} & 0\\ N_{13}^{T} & 0 & N_{33} \end{array}\right]<0 \end{array}$$

*where*

$$\begin{array}{c} \begin{array}{ccc} {\tilde{N}}_{11}\left(\nu \right) & := & \nu \left[A^{T}+A-\sigma^{2}\left(1-\nu \right)I\right],\\ N_{13}\left(\Lambda \right) & := & \alpha \sigma C^{T}\Lambda \Delta ,\\ N_{33}\left(\Lambda \right) & := & -\alpha^{2}\Delta \Lambda \end{array} \end{array}$$

*and α is a scalar that can be found using line search.*


**Remark 4.** 
*Note that if the conditions of Theorem 2 are fulfilled, then the solution x(t)≡0 is also exponentially p-stable for p=ν, as detailed in the Appendix A.*


## 7. Numerical Examples

In this section, two numerical examples illustrating the previous theoretical results will be presented. The first corresponds to the case when no nonlinearity is present in System (Equation 4). The second is an application to a chaos model of the form System (Equation 4) with a nonlinearity.

**Example 1.** 
*Consider an open-loop unstable system of the form System (Equation 4) with F=0, with a natural frequency of 5 rad/s and damping coefficient of ζ=−0.01. The dynamic matrix of this system is*

$$A=\left[\begin{array}{cc} 0.1 & -6.25\\ 4 & 0 \end{array}\right]$$

*One can see that in this case, $\bar{\lambda }\left(A+A^{T}\right)=2.3522$. We consider a couple of values of the driving noise intensity, corresponding to σ=0.5 and σ=2. For σ=2, Condition (Equation 6) from Theorem 1 is fulfilled and therefore the solution x(t)≡0 is asymptotically stable in probability. This conclusion is illustrated in Figure 2, which presents the time responses of the states of the system. It can be shown that the case with too small σ fails to achieve SAR whereas the case with large enough σ achieves SAR.*


**Example 2.** 
*We next consider a slightly modified version of the third-order chaos generator model of [25] with a single nonlinearity, described by System (Equation 4) and*

(21)
$$\begin{array}{c} A=\left[\begin{array}{ccc} -\epsilon & 1 & 0\\ 0 & -\epsilon \; & 1\\ a_{1} & a_{2} & a_{3} \end{array}\right],\;F_{1}=\left[\begin{array}{c} 0\\ 0\\ 10 \end{array}\right],\;C_{1}^{T}=\left[\begin{array}{c} \beta \\ 0\\ 0 \end{array}\right],\\ D=\sigma I_{3}, \end{array}$$

*where $a_{1}=-2,a_{2}=-1.48,a_{3}=-1,\epsilon =0.01$, and β=1. The nonlinearity is $f\left(y_{1}\right)=tanh\left(y_{1}\right)$. In order to apply the result of Theorem 2 together with its assumptions, one defined $F=\left[\begin{array}{cc} F_{1} & 0_{3\times 2} \end{array}\right]$, $C=I_{3}$, S=diag(1,0,0), and Δ=diag(1,0,0). Checking for different values of ν∈(0,1) and σ>0, the feasibility of Inequality (Equation 10), which is linear with respect to the variables *Λ* and T, one obtains that, for instance, for ν=0.2 and σ=3, the conditions of Theorem 2 for asymptotic stability in probability of the considered system are accomplished by Λ=diag(0.0333,0.1279,0.0803) and T=diag(1.0018,0.6471,0.6388), in which we used [26] to solve the linear matrix inequalities from the statement of Theorem 2.*


Next, we simulate the above system for 1000 s with an integration step of 0.001 s with σ=0 for t≤500 s and σ=3 for the rest of the time. The results are given in Figure 3 and Figure 4. The phase-plane (i.e., $x_{1}$ versus $x_{2}$) trajectories are depicted in Figure 3, and the components $x_{i},i=1,2,3$ of the state vector as a function of time are depicted in Figure 4. It can be seen from these figures that the chaotic behavior characterizing the system without the state-multiplicative noise is replaced by a stable trajectory at t≥500 s, in which SAR is attained. Thus, the feasibility of stabilization by multiplicative noise is demonstrated, using the stochastic control input u(t)dt:=σx(t)dβ(t).

## 8. Conclusions

The phenomenon of Stochastic Antiresonance (SAR) for a class of systems with sector-bounded nonlinearities has been considered. The stochastic stability is analyzed using a specific non-quadratic version of a Lur’e-type function for the considered application. This analysis leads to sufficient conditions for stability that are expressed as Linear Matrix Inequalities (LMIs), which, in turn, can be solved using standard convex optimization packages. Using those LMIs, one can determine the intensity of the state-multiplicative noise that stabilizes a noise-free unstable system, including systems that exhibit a chaotic behavior. One such system has been numerically simulated in which it has been shown that stability is attained (i.e., oscillations decay) shortly after the onset of the state-multiplicative noise, namely achieving SAR. The intensity of the applied noise that achieves SAR is in accordance to the LMI conditions. The considered class of systems with sector-bounded nonlinearities correspond to a large number of practical applications. However, although the presented developments do not yet comply with biological neuron models, the success in stability analysis and SAR demonstration encourages further research to treat more general models, such as [3], and more complex ones, on the way of using noise in neuromodulation and other applications, serving as a means of control rather than as a destructive effect. The possible modeling approach for such systems, may apply approximate replacement of continuous nonlinearities not complying with the sector conditions with function-fitting neural networks of a single hidden layer through a sector-bounded activation function and a linear output layer. The latter modeling approach is left as a topic for future research. One may also consider exploring possible SAR in other fields, e.g., aero-elastic systems, where Stochastic Resonance emerges as a response to additive noise [27]. Subsequent developments may be dedicated to the case when the linear terms of System (Equation 4) are replaced by nonlinear functions satisfying some smoothness, boundedness, and commutation assumptions, as considered in [28]. Furthermore, alternative stability analysis approaches can be used, using, for instance, the almost global stochastic stability conditions derived in [29].

## Figures and Tables

**Figure 1 entropy-26-00115-f001:**
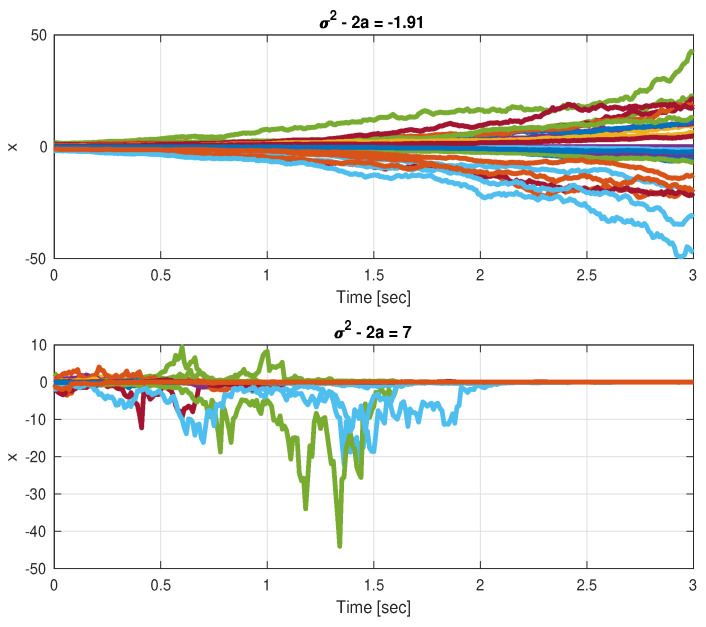
A scalar system that is unstable without state-multiplicative noise, (**top**) $\sigma^{2}<2a$ SAR not attained, (**bottom**) $\sigma^{2}>2a$ SAR attained.

**Figure 2 entropy-26-00115-f002:**
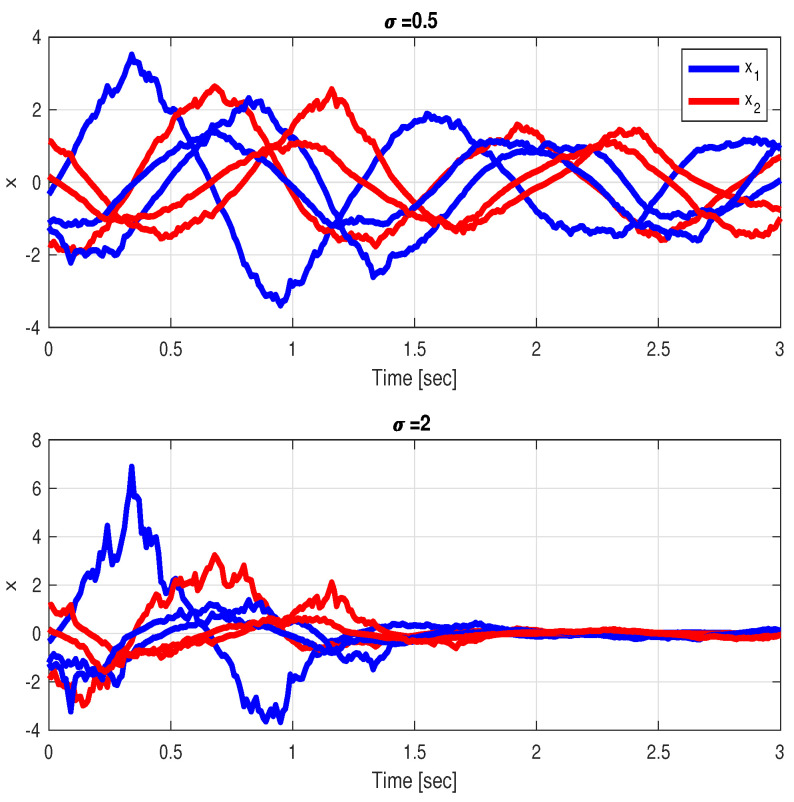
Second-order system subject to state-multiplicative noises of different intensities, (**top**) For σ=0.5, $\bar{\lambda }\left(A+A^{T}\right)>\sigma^{2}$, SAR not attained, (**bottom**) For σ=2, $\bar{\lambda }\left(A+A^{T}\right)<\sigma^{2}$, SAR attained.

**Figure 3 entropy-26-00115-f003:**
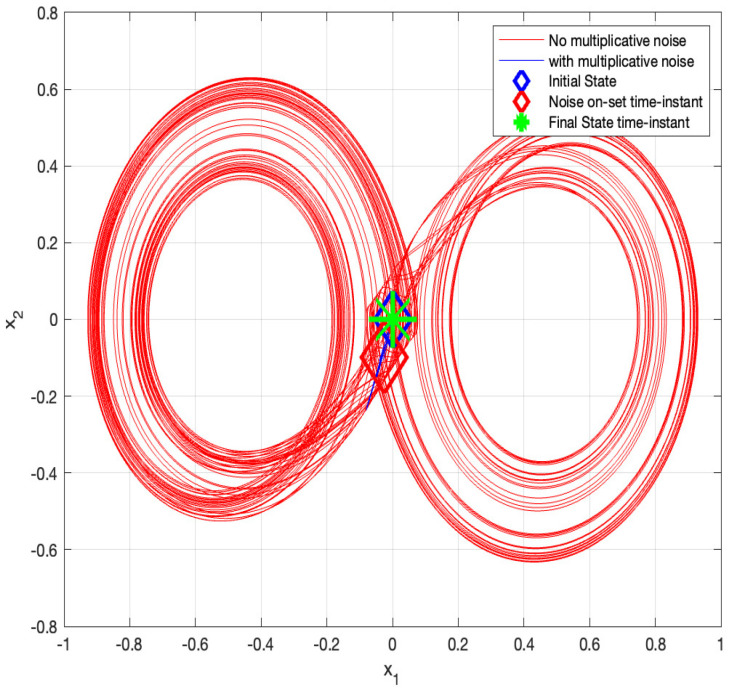
$x_{1}$ vs. $x_{2}$: Chaos stabilization using multiplicative noise, the original σ=0 jumps to σ=3 at t=500 s.

**Figure 4 entropy-26-00115-f004:**
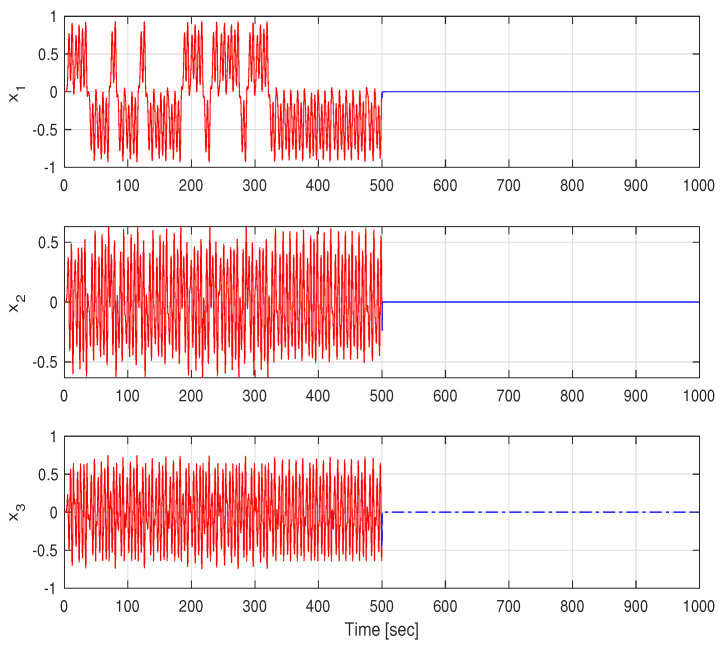
States time responses before and after applying the state-multiplicative noise.

## Data Availability

The paper contains all the data needed for reproducting the presented numerical results.

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
