# Peer review of "Stochastic Antiresonance for Systems with Multiplicative Noise and Sector-Type Nonlinearities"

_entropy, 2024, doi:10.3390/e26020115_

Round 1
Reviewer 1 Report
Comments and Suggestions for Authors
This paper studies the phenomenon of stochastic anti-resonance. In particullar, they derive a condition for a class of systems under which systems will have this behavior.
The results are to the best of my knowledge new, correct and interesting. Therefore, I recommend publication in Entropy.
I do have a few small suggestions that the authors might want to take into account prior to publication:
- the sentence 'The larger the value of x, so are the corrections larger' seems grammatically incorrect.
- Between eq. 4 and Eq. 5, the authors write 'y_i=C_ix'. It is not clear to me what C_i is. Shouldn't this be \sum_{j}C_{ij}x_j?
- It would be interesting to see a discussion of the direct physical implications of the results, for example, by looking at a physical example in section7.
Comments on the Quality of English Language/
Reviewer 2 Report
Comments and Suggestions for Authors
See attached review.
